# OpenReview forum: "Watch the Weights: Unsupervised monitoring and control of fine-tuned LLMs"
_NeurIPS.cc/2025/Workshop/Reliable_ML — NeurIPS 2025 - Reliable ML Workshop_

### Official Review · Reviewer_W2kD · 2025-09-19
**Accept**

**Rating:** 7
**Confidence:** 3

**Review:**

The paper proposes WEIGHTWATCH, a data-free method for auditing and monitoring fine-tuned LLMs by analyzing weight differences ΔW between a fine-tuned model and its base. The top singular vectors of ΔW are treated as behavioral directions; monitoring uses cosine similarity of activations to these directions, and steering removes anomalous components to nudge behavior back toward baseline. Reported results include up to 100% backdoor blocking with <1.2% FPR, detection of “unlearned” content up to 95.42%, and qualitative audits of instruction-tuned models (OLMo, Llama, Qwen) revealing fine-tuning priorities.

## Strengths:
- Practical reliability impact: Works without training-data access, directly addressing auditing under imperfect/unknown data regimes.
- Simple, general mechanism: SVD on ΔW with per-direction monitoring/steering; clearly framed and easy to adopt.
- Strong results: High detection and mitigation for backdoors, meaningful unlearning verification, and informative real-model audits.

## Weaknesses:
- Base-model dependency: Access to base weights is assumed; feasibility without them isn’t evaluated. (Method centers on ΔW.)
- Coverage limits: Behaviors not well aligned with top ΔW directions may evade detection; sensitivity to number of directions (k) and thresholds is not fully characterized.
- Operational overhead: The paper could quantify latency/throughput overheads when monitoring many directions across layers.

## Suggestions:
- Report ROC/AUPRC and overhead as a function of k (directions) and thresholding strategy; offer a default deployment recipe.
- Stress-test against stealthy or dispersed backdoors and gradient-obfuscated fine-tunes; include negative results.
- Clarify steering safeguards (policy gating, audit logs) and interactions with safety layers.
- Discuss scenarios without base-model access (e.g., approximate base recovery or synthetic anchors).

## Ethics:
Positive: enables pre-deployment auditing and low-FPR monitoring that can reduce harm from poisoned data/models. Risks: steering/jailbreak capability could undo safety unlearning; release should include acceptable-use policies and guardrails.

---

### Official Review · Reviewer_viVi · 2025-09-19
**solid paper, interesting claims!**

**Rating:** 9
**Confidence:** 5

**Review:**

This paper introduces a method for detecting backdooring and malicious fine-tuning of of open-weights models by inspecting the difference between the base models weights and the fine-tuned weights.


The weakness of this method is assuming you have access to both sets of weights. In practice, this might be too strong of an assumption.  However, if having access to both weights yields strong security benefits, that would be a compelling reason to advocate for open-weights models.

Alg.

1. compute weight difference between linear layers in transformers -> \delta W
2. SVD this matrix \delta W
3. get top singular vectors of this


4 At inference, compute Cosine Similarity between latent activations and the directional vectors from 3 these using these
5 "Large" values correspond to anomalies

The authors conduct relatively detailed experiments on various fine-tuning based attacks (malicious PPO,LoRA, and FFT) and some in the wild results (applying their method to real world LLM fine-tunes). These in the wild results seem to deduce facts about the fine-tuning data mix of these  models that are confirmed by the papers that released the models (especially the data-mix for OLMO).

In general, the method seems promising. I think the focus on weights vs activations (which need to be computed using a specific reference data-set) is compelling. Hard to know if this will be a durable defense, but the in the wild results are intriguing.


Clearly passes the bar for workshop paper.

My only comment to the authors

1. would be to make explicit the contrast with prior methods---like expand figure 1
2. maybe conduct an adaptive attack on the defense?

---

### Official Review · Reviewer_Wfav · 2025-09-23
**This paper proposes WEIGHTWATCH, a method for monitoring, interpreting, and controlling fine-tuned LLMs by analyzing weight differences.**

**Rating:** 7
**Confidence:** 4

**Review:**

Strengths

- The shift of focus from activations to weight differences is well-motivated, as it offers a clean and scalable alternative that avoids reliance on training data.

- The experimental results are very strong. Particularly, "in-the-wild" analysis of the method shows that the method proposed in the paper has potential to be impactful in model auditing.

Weaknesses

- One obvious weakness is the dependence on base model weights. Now-a-days models undergo many stages of training, including many stages of fine-tuning. The paper does not talk about the performance of the method assuming models after different stages as base models.

- Almost all the ideas are already seen in the model merging literature. Even though authors write a paragraph in the appendix about model merging, it is underdone. It needs to be in the main paper and more rigorous analysis of the model merging literature, exactly identifying what this paper is achieving.